# Synthesis of Nitro- and Acetyl Derivatives of 3,7,10-Trioxo-2,4,6,8,9,11-hexaaza[3.3.3]propellane

**DOI:** 10.3390/ma15238320

**Published:** 2022-11-23

**Authors:** Vera S. Glukhacheva, Sergey G. Il’yasov, Elena O. Shestakova, Egor E. Zhukov, Dmitri S. Il’yasov, Anastasia A. Minakova, Ilia V. Eltsov, Andrey A. Nefedov, Alexander M. Genaev

**Affiliations:** 1Institute for Problems of Chemical and Energetic Technologies, Siberian Branch of the Russian Academy of Sciences (IPCET SB RAS), Biysk 659322, Russia; 2Department of Organic Chemistry, Faculty of Natural Sciences, Novosibirsk State University, Novosibirsk 630090, Russia; 3Department of Physical Organic Chemistry (OPOC), Novosibirsk Institute of Organic Chemistry, Siberian Branch of the Russian Academy of Sciences (NIOCh SB RAS), Novosibirsk 630090, Russia

**Keywords:** propellane, nitration, acetylation, quantum-chemical predictions

## Abstract

Here, we report the study results of the nitration of 3,7,10-trioxo-2,4,6,8,9,11-hexaaza[3.3.3]propellane (THAP) by different nitrating agents such as nitric acid, mixed nitric/sulfuric acids, nitric anhydride, and mixed concentrated nitric acid/acetic anhydride to furnish 3,7,10-trioxo-2-nitro-2,4,6,8,9,11-hexaaza[3.3.3]propellane and 3,7,10-trioxo-2,8-dinitro-2,4,6,8,9,11-hexaaza[3.3.3]propellane, whereas a lactam–lactim rearrangement was found to take place upon vigorous cooling to give 10-hydroxy-2,4,6,8,9,11-hexaazatricyclo[3.3.3.01,5]undec-9-ene-3,7-dione. The two competing reactions, lactam–lactim rearrangement, and nitration were found to take place. The acylation of 3,7,10-trioxo-2,4,6,8,9,11-hexaaza[3.3.3]propellane was examined and the formation conditions of 2,6-di- and 2,6,9-triacetyl-substituted and 3,7,10-trioxo-2,4,6,8,9,11-hexaacetyl-2,4,6,8,9,11-hexaaza[3.3.3]propellane were established. The acetyl derivatives were found to be instable in an acidic medium and to undergo deacylation. The obtained findings correlate well with the quantum-chemical calculations.

## 1. Introduction

Propellanes are an interesting class of compounds that embraces a wide range of not only natural but also synthetic origins. It is because of the high molecular density that they are among the promising caged organic compounds [1]. From among [3.3.3]propellanes, of great interest are heterocyclic compounds, especially those having nitrogen atoms (aza propellanes) as they are easily functionalizable. The nitro derivatives of aza[3.3.3]propellanes hold promise as compounds with high energy. The literature reports estimations of the energetic characteristics of some hypothetical nitro derivatives of aza[3.3.3]propellanes (Figure 1, Table 1) [2,3].

It can be seen from the prediction data that the density, detonation rate and detonation pressure of 2,4,6,8,9,11-hexanitro-2,4,6,8,9,11-hexaaza[3.3.3]propellane (**3**, HNHAP) are superior to those of the common standard explosives and almost similar to those of CL-20, the most powerful explosive at present, whose main drawback is known to be its high synthesis cost, impeding its wide application. Therefore, the quest for new, more affordable high-energy compounds with similar properties is being continued worldwide.

Shin et al. [4] reported data on the nitration of 3,7,9,11-tetraoxo-2,4,6,8,10-pentaaza[3.3.3]propellane **12** leading to 2,6-dinitro- (**13**) and 2,6,10-trinitro-3,7,9,11-tetraoxo-2,4,6,8,10-pentaaza[3.3.3]propellane (**14**). That said, the synthesis of a pentanitro-substituted 3,7,9,11-tetraoxo-2,4,6,8,10-pentaaza[3.3.3]propellane (**15**) failed (Figure 1).

In another study, Lee et al. [5] reported a synthetic method for the [3.3.3]hexaazapropellane cage bearing six NH groups, 3,7,10-trioxo-2,4,6,8,9,11-hexaaza[3.3.3]propellane (**19**, THAP) (Figure 2).

Compound **19** is composed of three imidazolidine rings (Figure 2). The structural precursors of **19** are imidazolidin-2-one (ethylene urea) and 2,4,6,8-tetraazabicyclo [3.3.0.]octane-3,7-dione (glycoluril), which structurally contain one or two imidazole rings, respectively. The nitration reactions of these structural precursors of THAP with mixed nitric/sulfuric acids, nitric acid, acetyl nitrate, or mixed ammonium nitrate/trifluoroacetic anhydride in nitromethane and with nitronium trifluoromethanesulfonate in dichloromethane are well-studied [6,7,8,9,10].

Despite the theoretical predictions of the nitro derivatives of [3.3.3]propellanes and the prospects for using the same as energetic compounds, there currently exists a major gap in the synthetic methodology of both aza[3.3.3]propellanes themselves and the high-energy derivatives thereof. The literature has, however, only a single report on the nitration of 3,7,10-trioxo-2,4,6,8,9,11-hexaaza[3.3.3]propellane with nitric acid in trifluoroacetic acid to 3,7,10-trioxo-2,8-dinitro-2,4,6,8,9,11-hexaaza[3.3.3]propellane [11].

Therefore, the present study aimed to explore the reaction of compound **19** with different nitrating systems such as nitric acid, mixed nitric/sulfuric acids, nitric anhydride (N_2_O_5_), and mixed nitric acid/acetic anhydride.

## 2. Materials and Methods

*General information*. Infrared spectra of the samples were recorded in KBr on a FT-801 Fourier spectrometer (Simex, Russia) at 4000 to 500 cm^−1^. Elemental analysis was performed on a CHNO FlashEATM 1112 analyzer. The melting point was measured on a Böetius PHMK instrument (Veb Analytik, Dresden). The decomposition temperature was measured on TGA/SDTA 851e and DSC 822e thermal analyzers (Mettler Toledo, Greifensee, Switzerland) over temperature ranges of 25–300 °C under nitrogen at a heating rate of 10 °C/min. The results were digitized and processed in STARe 11.0 thermal analysis software. Melting points were measured on a Stuart SMP30 (Stuart, Chelmsford, UK) digital melting point apparatus and on an instrument for the measurement of boiling points and melting in the capillary.

The NMR spectra were recorded in DMSO-d_6_, acetonitrile-d_3_, and methanol-d_4_ solutions (40 mg in 0.6 mL) at room temperature on Bruker Avance III 500 and Bruker AM-400 spectrometers (Billerica, MA, USA). The operating frequencies for ^1^H, ^13^C and ^15^N were 500.03, 125.73, and 50.67 MHz for Bruker Avance III 500, and 400.13 and 100.61 for Bruker AM-400. Chemical shifts of the signals are given on the δ-scale. Signals form the solvents were used as the standard for ^1^H and ^13^C NMR spectra. ^15^N NMR spectra were taken relative to formamide as the external standard (δ (^15^N) = 112.5 ppm). Signal assignment was performed using 2-D (HMBC) NMR spectroscopy.

Mass spectra and precise measurements of molecular weights were done on a Thermo Electron Double Focusing System (Thermo Electron Scientific Instruments Corporation, Madison, WI, USA). The samples contained in metal vials were introduced into the mass spectrometer by direct injection; if necessary, the sample vial can be heated up in a temperature range from 25 to 360 °C. The mass spectrometer was operated in the electron ionization mode at an electron energy of 70 eV. Measurements of the exact masses of ions were performed with respect to the standard lines of perfluorokerosene (PFK). Measurements of the exact masses of some compounds (**20** and **21**) were performed in solution on a Bruker MicroTOF-Q liquid mass spectrometer (Bruker Corp., Billerica, MA, USA), with the electrospray ionization method.

The progress of the reactions described herein and the purity of the synthesized products were monitored by the TLC method on Merck Silica gel 60 F_254_ plates with eluent selection, UV radiation or iodine vapor. The reaction products were separated and isolated using column chromatography on silica gel.

The chemicals—benzyl chloride, benzyl bromide, methyl iodide, ethyl bromide, propyl iodide, tert-butyl bromide, isopropyl chloride, trimethylamine, N,N-dimethylaminopyridine, sodium persulfate, trifluoroacetic acid, and acetyl chloride—were purchased from Acros. Uric acid was purchased from Angene Chemical. Acetic anhydride, chloroform, DMSO, DMF, 25% aqueous ammonia, benzylamine, tert-butylamine, isopropylamine, methylamine, and trimethylamine were purchased from Reachim. Mixed acid (48/52) was prepared by mixing conc. nitric acid (98%) with sulfuric acid (96%) and technical-grade oleum (24%).

The synthesis of THAP was performed by the reported procedure [5].

### 2.1. Synthesis of 10-Hydroxy-2,4,6,8,9,11-hexaazatricyclo[3.3.3.01,5]undec-9-ene-3,7-dione (20)

To concentrated nitric acid (7 mL) cooled to 0 °C was added THAP (0.5 g). The reaction mixture was then held at −40 °C and stirred for 1 h. After that, the mixture was poured into ice. Precipitated white crystals were collected by filtration and dried. Yield: 86%. IR, cm^−1^: 3252, 1804, 1752, 1701, 1496, 1466, 1393, 1191, 1162, 1133, 1094, 1050, 999, 778. ^1^H NMR (500 MHz, DMSO) δ 8.72 (s, 2H, 4,9-NH), 8.85 (s, 2H, 2,11-NH), 10.41 (s, 1H, 6-NH). ^13^C NMR (126 MHz, DMSO) δ 82.45 (1-C), 82.92 (5-C), 149.95 (7-C-OH), 158.10 (3,10-C=O). ^15^N NMR (51 MHz, HCONH_2_) δ 105 (2,11-N, ^1^J_NH_ = 97 Hz), 110 (4,9-N, ^1^J_NH_ = 95 Hz), 118 (6-N, ^1^J_NH_ = 95 Hz), 281 (C=N). Calcd for C_5_H_6_N_6_O_3_ [M]^+^ 199.0572; found *m/z* 199.051.

### 2.2. Synthesis of 3,7,10-Trioxo-2-nitro-2,4,6,8,9,11-hexaaza[3.3.3]propellane (21)

To concentrated nitric acid (5 mL) cooled to 0 °C was added THAP (0.3 g). The reaction mixture was then held at 0–10 °C and stirred for 1 h; afterwards, it was poured into ice and dried. Precipitated white crystals were collected by filtration and dried. Yield: 79%. Mp: becomes charred above 300 °C. IR, cm^−1^: 3406, 3238, 1814, 1751, 1697, 1554, 1485, 1441, 1390, 1333, 1258, 1185, 1155, 1091, 951. ^1^H NMR (500 MHz, DMSO) δ 8.80 (br. s, 2H, 6,9-H), 9.03 (br. s, 2H, 8,11-H), 10.17 (s, 1H, 4-H).^13^C NMR (126 MHz, DMSO) δ 80.39 (octet, *J*_CH_ = 2.6 Hz, 1-C), 86.50 (octet, *J*_CH_ = 2.6 Hz, 5-C), 146.60 (d, *J*_CH_ = 4.7 Hz, 3-C), 158.19 (t, ^2^*J*_CH_ = 3.4 Hz, 7,10-C). ^15^N NMR (51 MHz, HCONH_2_) δ 106 (1J_NH_ = 96.7 Hz, 8,11-N), 111 (1J_NH_ = 94.4 Hz, 6,9-N), 115 (1J_NH_ = 95.2 Hz, 4-N), 219 (2-N). Calcd. for C_5_H_5_N_7_O_5_ [M+NH_4_]^+^ 261.0691; found *m/z* 261.053.

### 2.3. Synthesis of 3,7,10-Trioxo-2,8-dinitro-2,4,6,8,9,11-hexaaza[3.3.3]propellane (22)

To mixed nitric/sulfuric acids (48:52) pre-cooled to 0 °C was added THAP (0.3 g). The reaction mixture was then stirred for 1 h at 23 °C; afterwards, it was poured into ice. The white precipitate was collected by filtration and dried. Yield: 0.26 g (59.9% of the theor.). Mp: 209 °C. IR, cm^−1^: 3345, 3102, 2835, 1806, 1745, 1582, 1453, 1409, 1332, 1264, 1178, 1083. ^1^H NMR (500 MHz, DMSO) δ 9.66 (s, 2H, NH-CO-NH), 10.87 (s, 2H, NH-CO-N-NO_2_).^13^C NMR (126 MHz, DMSO) δ 81.27 (t, ^2^*J*_CH_ = 3.9 Hz, C(N)_3_), 145.89 (s, CO(N-NO_2_)), 157.27 (t, ^2^*J*_CH_ = 3.4 Hz, NH-CO-NH). ^15^N NMR (51 MHz, HCONH_2_) δ 105.08 (NH-CO-NH, ^1^*J*_NH_ = 48 Hz). HR-MS: calcd. for C_5_H_4_N_8_O_7_ [M]^+×^ 288.0198; found *m/z* 288.0196.

### 2.4. Nitration of 21 and 22 with Mixed Nitric Acid/Acetic Anhydride (23)

To concentrated nitric acid (8 mL) cooled to 0 °C was added **21** or **22** (0.3 g). Acetic anhydride (3.6 mL) was then slowly added portionwise, and the reaction mixture was held at room temperature and stirred for 2 h. Afterwards, the reaction mixture was cooled to 2 °C, and the precipitated crystals were collected by filtration, washed with methylene chloride, and dried. Yield: 79%. Mp: 146 °C. IR, cm^−1^: 3399, 3111, 1803, 1630, 1599, 1402, 1317, 1165, 881. ^1^H NMR (400 MHz, DMSO) δ 11.22 (s, 2H, 4,6-H), 11.78 (s, 1H, 1-H). ^13^C NMR (100 MHz, DMSO) δ 150.14, 154.25, 159.63. Calcd. for C_5_H_3_O_9_N_9_ [M]^+^ m/z = 333; found *m/z* = 286.0774 and 195.0488.

### 2.5. Acetylation of 3,7,10-Trioxo-2,4,6,8,9,11-hexaaza[3.3.3]propellane

3,7,10-trioxo-2,4,6,8,9,11-hexaaza[3.3.3]propellane (0.3 g, 0.0015 mol) and concentrated sulfuric acid (0.3 mL, 0.006 mol) were added to acetic anhydride (15 mL). The reaction mixture was heated to 90 °C and held at this temperature for 2 h. Afterwards, the sediment was collected by filtrated and acetic anhydride evaporated. Both sediments were combined and dissolved in hot water. The undissolved sediment was compound **24**. Yield: 0.33 g (68% of the theor). IR, cm^−1^: 3235, 2943, 2824, 1775, 1722, 1677, 1471, 1382, 1326, 1180, 1113, 1096, 993. ^1^H NMR (500 MHz, DMSO) δ 2.36 (s, 6H, 2,9-CH_3_), 8.67 (s, 1H, 6,8-NH), 9.74 (s, 2H, 4,11-NH). ^13^C NMR (126 MHz, DMSO) δ 23.15(2,9-CH_3_), 82.40 (1,5-C), 151.98 (3,10-C=O), 157.91 (7-C=O), 168.52 (2,9-C=O). ^14^N NMR (51 MHz, HCONH_2_) δ 107,114,173. Calcd. for C_11_H_12_N_6_O_6_ (%): C, 40.71; H, 3.70; N, 25.91; found (%): C, 40.62; H, 3.46; N, 25.74. Calcd. for C_11_H_12_O_6_N_6_ [M]^+^
*m/z* = 324.0813; found *m/z* = 324.0810.

Water removal resulted in compound **25**. Yield: 0.37 g (80%). IR, cm^−1^: 3292, 2941, 2831, 1758, 1722, 1693, 1453, 1423, 1377, 1321, 1185, 1134, 1045, 980, 844. ^1^H NMR (500 MHz, DMSO) δ 2.34 (s, 6H, 2,9-CH_3_), 2.36 (s, 3H, 8-CH_3_), 9.59 (s, 1H, 11-NH), 9.99 (s, 2H, 4,6-NH). ^13^C (126 MHz, DMSO) δ 23.15 (CH_3_), 82.40 (C_quater._), 151.98 (C=O), 157.91 (C=O), 168.52 (CH_3_-C=O). ^15^N NMR (51 MHz, HCONH_2_) δ 108 (11-N, 1J_NH_ = 98 Hz), 111 (4,6-N, 1J_NH_ = 98 Hz), 163 (2,8-N), 169 (4-N). Calcd. for C_11_H_12_O_6_N_6_ [M]^+^ m/z = 324.0813, found m/z = 324.0810.

### 2.6. Synthesis of 3,7,10-Trioxo-2,4,6,8,9,11-hexaacetyl-2,4,6,8,9,11-hexaaza[3.3.3]propellane (26)

3,7,10-trioxo-2,6-diacetyl-2,4,6,8,9,11-hexaazaacetyl-2,4,6,8,9,11-hexaaza[3.3.3]propellane (0.28 g) and acetyl chloride (4.3 mL) were added to methylene chloride (15 mL), and the whole mixture was stirred for 20 min at room temperature. Triethylamine (0.73 mL) was then added though a dropping funnel, and the mixture was heated to boil for 2 h. Upon completion of the reaction, the stock solution was evaporated to give a white sediment, which was washed with water and dried. Yield: 0.9 g (20%). IR, cm^−1^:2927, 1766, 1721, 1659, 1619, 1400, 1371, 1253, 1190, 1036, 978. ^1^H NMR (400 MHz, DMSO-*d*6) 2,37 (18H, s, CH_3_). ^13^C NMR (100 MHz, DMSO) δ 23.74 (CH_3_), 82.91 (C_quater._), 152.54 (C=O), 169.17 (CH_3_-C=O). Calcd. for C_17_H_18_O_9_N_6_ [M]^+^
*m/z* = 450.3596; found *m/z* = 450.3591.

### 2.7. Synthesis of 3,7,10-Trioxo-2,6-diacetyl-9-nitro-2,4,6,8,9,11-hexaaza[3.3.3]propellane (27)

To conc. HNO_3_ (3.3 mL) cooled to 0 °C was added 3,7,10-trioxo-2,6-diacetyl-2,4,6,8,9,11-hexaaza[3.3.3]propellane (18 g, 0.6 mmol). Then, acetic anhydride (2.3 mL) was added from the dropping funnel in such a way that the temperature does not rise above +15 °C. The reaction mixture was held at +25 °C for 1 h. Upon completion of the holding time, the reaction mixture was poured into ice to generate a sediment. The sediment was collected by filtration, washed with water and acetone, and dried under atmospheric pressure. Yield: 0.13 g (57%). Mp: 241 °C. IR, cm^−1^: 3372, 3235, 1778, 1677, 1588, 1448, 1382, 1325, 1179, 1157. ^1^H NMR (500 MHz, DMSO-*d*6) 2.38 (3H, s, CH_3_), 2.40 (3H, s, CH_3_), 10.29 (1H, s, NH), 10.39 (1H, s, NH), 10.56 (1H, s, NH), ^13^C NMR (126 MHz, DMSO-*d*6) 23.80 (CH_3_), 24.46 (CH_3_), 80.61 (C_quater._), 83.63 (C_quater._), 148.93 (C=O), 150.85 (C=O), 151.29 (C=O), 168.58 (C=O), 169.49 (C=O), ^15^N NMR (51 MHz, HCONH_2_) δ 108, 112, 170, 214. Calcd. for C_11_H_12_N_6_O_6_ (%): C, 40.71; H, 3.70; N, 25.91; found (%): C, 40.62; H, 3.46; N, 25.74. HR-MS: calcd. for C_9_H_9_O_7_N_7_ [M]^+×^
*m/z* = 327.0558; found *m/z* = 327.0554.

Quantum chemical predictions: The calculations were performed using a fast DFT code implemented in the PRIRODA program [12,13,14] and employing a PBE functional [15,16,17,18] with full-electron basis Λ0120,22 (similar to the cc-pVDZ basis) as a gas-phase model. Conformational analysis was performed with the MarvinSketch program using conformers plugin15 and with the CREST program23 using GFN2-xTB24 theory. The geometries of all the conformers were then optimized on DFT/PBE/Λ01 level. The heat of formation of the compound CcHhNnOo was calculated by the formula ΔfH0 = (E + 38.0843685585c + 0.5872542349h + 54.7271355293n + 75.1115133857o)627.51 (kcal/mol) where E is TOTAL ENERGY for the DFT/PBE/Λ01 method (a.e.). The coefficients c, h, n, o in this formula were obtained according to the method [http://openmopac.net/manual/SCF_calc_hof.html (accessed on 18 November 2022)] by minimizing the least-squares deviations of the calculated energies from the experimental heats of formation using the test set [http://openmopac.net/PM7_and_PM6-D3H4 accuracy/table_of_heats.html (accessed on 18 November 2022)], limited to neutral molecules containing elements C, H, N, O only (about 1000 compounds) (see Appendix A).

## 3. Results and Discussion

The synthesis of nitro derivatives of saturated nitrogen-bearing cycles is quite difficult and accompanied by a good deal of reaction products. Therefore, it is important to search for approaches that enhance the selectivity and effectiveness of these processes.

The positive induction effect of the adjacent CH and CH_2_ groups favors an electrophilic substitution reaction. Therefore, ethylene urea is nitrated in one stage, while a complete substitution in glycolurils requires two nitration stages. It can thus be speculated that the presence of the tertiary carbon atom, as well as the bulky tricyclic structure, hinder a complete substitution, and the subsequent nitration of the THAP nitro derivatives should be carried out under harsh conditions.

Our study into the effects of different nitrating agents began with 98% nitric acid (Figure 3, Table 2).

It is seen (Table 2) that the composition of the reaction products depends on the reaction temperature. For instance, the reaction product was compound **20** when temperature was −40 °C. The ^1^H NMR spectrum of the sample shows three signals from the amino groups at 8.74, 8.87, and 10.42 ppm with an intensity ratio of 2:2:1. The ^13^C NMR spectrum of the test compound shows two signals from the inequivalent nodal atoms of carbon at 82.50 and 82.97 ppm, as well as two signals of carbon atoms relating to the urea moiety at 149.99 and 158.16 ppm. The analysis of 2D spectroscopy noted that the hydrogen atom at 10.42 ppm interacts with two carbon atoms (149.99 ppm and 82.97 ppm) and two nitrogen atoms (direct interaction of N-H, δ = 118 ppm, ^1^J_NH_ = 95 Hz, long-range interaction δ = 281 ppm). The hydrogen atom at 8.74 ppm interacts with the quaternary carbon atom at 82.97 ppm and the carbonyl carbon at 158.16 ppm. The ^1^H,^15^N HMBC spectrum shows a direct constant of ^1^J_NH_ = 95 Hz on the nitrogen atom located at 110 ppm and a cross peak on the other nitrogen atom at 105 ppm. A similar pattern is also observed for the last signal from ^1^H at 8.87 ppm: the intrinsic N-H coupling constant ^1^J_NH_ = 97 hz (δ = 105 ppm) and the cross peak with a nitrogen signal from the amino group δ = 110 ppm. This signal also interacts with the carbon signal at 158.16 ppm but produces a cross peak with the second nodal carbon atom at 82.50 ppm.

Overall, it can be said from the data presented that this sample retained the starting three-membered cage of THAP; however, one of the cycles underwent a change affecting one amino group as a result of the reaction. In the ^15^N NMR spectrum, we observed a well-detectable signal at 281 ppm. This position is typical of sp^2^-hybrid nitrogen atoms, evidencing the formation of an =N– bond. It can be speculated that the generation of such a structure is through the protonation of the carbonyl oxygen atom by the present acid, followed by stabilization of this form by the van der Waals interaction between the nitrogen atom and the hydroxyl hydrogen atom to furnish a lactim form of THAP **20**.

When the temperature was gradually raised, the portion of the lactim form of THAP was observed to decrease and mononitro derivative **21** was formed. The IR spectrum of this compound has a broad region of NH stretching vibrations. The absorption band at 1556 cm^−1^ is responsible for stretching vibrations of the nitro group. The presence of the nitro group in the ^1^H NMR spectrum is corroborated by the strong downfield shift of hydrogen atoms at 8.80 (br. s, 2H), 9.03 (br. s, 2H), and 10.17 (s, 1H) in a ratio of 2:2:1. The presence of only one nitro group in **21** resulted in the inequivalence of all hydrogen atoms and quaternary carbon atoms (^13^C NMR).

Starting from +10 °C, the insertion of the second nitro group (compound **22**) was detected, with a maximum yield achieved at +40 °C. The ^1^H NMR spectrum of the resultant compound **22** fits with the literature data [6]; two equally intense singlets are observed in the region above 9.5 ppm typical of aldehyde or mobile hydrogen atoms of the NH and OH types. The signal at 10.87 ppm is more broadened, suggestive of its high involvement in the exchange with water present in the system. The ^13^C NMR spectrum has three signals: one near 81 ppm and two near the carbonyl carbon atoms. It was found from the 2D spectroscopy that none of the signals in the ^1^H NMR spectrum had a direct coupling constant with the carbon atom. That said, two spin systems linked to each other via the carbon atom at 81 ppm are observed. It can thus be concluded that the molecule incorporates the following moieties. The first moiety is –NH-CO-NH–. Two equivalent hydrogen atoms and a carbon atom at 157 ppm. The nitrogen atom gives a signal at 105 ppm with the constant ^1^J_NH_ = 48 Hz, in good agreement with information on similar compounds. The next two moieties are –NH-CO-N(NO_2_)–. Likewise, the total number of hydrogen atoms equals the intensity of 2. Given that both quaternary carbon atoms give a single signal, i.e., they are equivalent between each other, the only option of the mutual arrangement is a structure wherein the nitro groups lie on the opposite side of the plane extending through three carbonyls.

Another reactant for the insertion of nitro groups into **19** was mixed 48:52 HNO_3_:H_2_SO_4_ (Table 3).

Unlike concentrated HNO_3_, this case showed a negligible formation of the lactim form, and starting from a temperature of +15–18 °C, 2,6-dinitro-THAP (**22**) became the major product with a maximum yield of 60% at room temperature. A further rise in temperature lowered the yield of product **22**.

Thus, the two competing reactions, lactam–lactim rearrangement, and nitration were observed to proceed in the course of nitration.

When nitric anhydride (N_2_O_5_) was used, the reaction products were the lactim form of THAP **20**, *trans*-isomer 2,6-dinitro-THAP (**22**), and *cis*-isomer 2,8-dinitro-THAP (**22^/^**) (Figure 4).

The samples that contained compound **22^/^** and had good solubility in deuterated DMSO appeared to be instable in solution, not allowing us to perform prolonged experiments. Shortly after the dissolution, the decomposition process began and was accompanied by gas liberation. The pot life of the samples was about 1.5 h. Significant signals relating to the preserved cage of THAP were the peaks near 80 ppm in the ^13^C NMR spectra.

The ^13^C NMR spectrum showed signals with close values of chemical shifts of compound **22** at 75.5, 87.7, 145.3, and 156.9 ppm. A weak cross peak at 10.78 ppm was observed together with a signal at 88 ppm in the ^1^H spectrum. It was assumed that those signals came from the other isomer **22^/^** with a *cis*-arrangement of the nitro groups. The *cis*-configuration of the nitro groups most likely destabilized the molecule and favored its fast decomposition. This is confirmed by the signals of quaternary carbon atoms disappearing in the spectrum over time, while *trans*-isomer **22** remained. The spectral pattern typical for compound **22^/^** changed in 1.5 h, pointing to the decay products of the molecule (Figure 2).

The synthesis processes of the hexanitro derivative of THAP were complicated by the formation of **21** and **22** and by the reduced basicity of the NH group to increase the number of nitro groups. Therefore, when selecting a nitrating agent for further nitration of 2-nitro- and 2,6-dinitro-THAP, we were oriented towards the synthetic process for 2,4,6,8-tetranitroglycoluril via 2,6-dinitroglycoluril and used mixed concentrated nitric acid and acetic anhydride as the nitrating agent [9,10].

The IR spectra of the compounds obtained by nitration of **21** and **22** are identical. The IR spectrum of the reaction product **23** shows stretching vibrations at 3399 and 3111 cm^−1^, typical of the NH groups, allowing for the assumption that nitration was incomplete.

***CAUTION!*** When preparing samples **23** for NMR spectroscopy, the addition of deuterated DMSO provoked an exothermal decomposition to give off NO₂ and a sound effect (a boom). The same happened when DMF and acetone were added.

The compound was insoluble in deuterated acetonitrile and H_2_O-*d*_2_, and the chemical shifts did not show up in the spectrum. The ^1^H NMR spectrum of the substance that decomposed in DMSO showed that the solution retained chemical shifts of the decay products of **22**.

The TGA and DSC results (Figure 3) demonstrate that the thermal stability is reduced by the incorporation of the nitro groups. For instance, compound **22** is more thermally stable, and a smooth stepwise decomposition by the temperature is observed, and an exothermic effect whose extremum is at 209 °C is recorded afterwards. Trinitro derivative **23** decomposes abruptly at 146 °C with a 83% weight loss. The thermal decomposition process self-accelerates when that temperature is achieved and transits consequently into a thermal explosion. The low thermal stability generally implies a high sensitivity to mechanical stimuli (a crackling noise can be heard when it is ground in an agate mortar).

Fragments of the compounds were detected in the mass spectrum of the resulting substance (Figure 4), but we failed to calculate the weight of the whole molecule because of the low thermal stability.

When **21** and **22** were nitrated with N_2_O_5_ and mixed acid, no further substitution occurred. The starting compounds were liberated from the reaction mixture.

Thus, because the resultant compound was instable in solution, we failed to acquire unambiguous data on the structure, but it can be speculated from the above-listed indirect signs that trinitro derivative **23** was formed (Figure 5).

The acetyl group is often utilized as a protecting group for amines [19] to ensure that the reaction is chemoselective, but in some instances, it is used to perform nitration, for example, in the nitrolysis of hexaazaacetylhexaazaisowurtzitane to hexanitrohexaazaisowurtzitane (CL-20) [20]. Therefore, a decision was made to explore the reactivity of compound **19** towards acetylation and chloroacetylation and examine whether nitro derivatives of **19** could be synthesized from its acetyl substituted ones.

Acetic anhydride was chosen as the acetylating agent for **19**. THAP is almost insoluble in acetic anhydride, even after an acid is added as the catalyst (sulfuric, phosphoric, and chloric) when heated. The reaction mixture represented a suspension throughout the entire reaction time. In the literature, acetylation is most frequently carried out by refluxing a reaction mixture (120–140 °C); [21]. However, a black residue of an uncertain composition was formed at this temperature in our case. When the temperature was lowered to 90 °C, the reaction proceeded within a few hours to yield acetyl derivatives (Figure 6, Table 4). A further decline in temperature required that the reaction time be extended, while the starting compound remained in the reaction mixture at room temperature.

The reaction products were the two compounds 2,6-diacetyl-THAP **24** and 2,6,9-triacetyl-THAP **25**. We managed to isolate the compounds individually because they had different solubility in some solvents; for instance, compound **24** was more soluble in water and ethyl acetate. It was also noted that increasing the acid content in the reaction mixture resulted in a decreased yield of 2,6,9-triacetyl-THAP, probably due to the acetyl derivatives of THAP being instable in the acidic medium, thereby leading to a counter reaction of deacetylation.

The IR spectra of compounds **24** and **25** have stretching bands of the associated NH groups in the region from 4000 cm^−1^ to 3000 cm^−1^. Three absorption bands from 1675 cm^−1^ to 1779 cm^−1^ can be attributed to C=O stretching vibrations.

The presence of the acetyl groups in the ^1^H NMR spectrum of compound **24** is corroborated by the signal present at 2.36 ppm typical of the hydrogen atoms of the methyl group of the acetyl moiety; moreover, the signal typical of the NH group proton is observed to shift downfield from 8.05 ppm to 9.74 ppm and 8.67 ppm. The presence of free protons and of protons of two acetyl groups is characterized by a peak ratio of 2:2:6, respectively.

The ^1^H NMR spectrum of product **25** shows overlapping signals from the methyls at 2.34 and 2.36 ppm, and two signals coming from the amino groups: broadened singlets at 9.99 ppm and 9.59 ppm at an intensity ratio of 1:2. In the ^13^C NMR, compound **25** gives two signals differing in intensity from the methyls at 23.19 ppm and 25.21 ppm, two signals from the nodal carbon atoms at 80.61 ppm and 84.05 ppm, two different-intensity signals from the carbon atoms relating to the urea moiety at 150.94 ppm and 152.74 ppm, and two different-intensity signals from the carbonyls of the acetyl moiety at 168.61 ppm and 168.85 ppm.

With regard to the addition of the three acetyl moieties, this situation is possible only in the case of monosubstitution of each five-membered ring. That said, the two rings must contain substituents on the same side of the plane going through the carbonyls of the cage, and the third ring must contain them on the opposite side.

The GC-MS spectra of **24–25** showed the presence of peaks of molecular ions [M]+ with a maximum intensity, whose *m/z* values match the calculated molecular weights of compounds of the proposed structure.

Among the efficient acetylation methods is reacting a substrate with acetyl chloride. A series of experiments were conducted to acetylate THAP with acetyl chloride in different solvents (DMSO, DMF, methylene chloride, 1,4-dioxane, toluene). However, no acetyl derivatives were formed, and the starting compound **19** was liberated in the same quantity.

The incorporation of acetyl groups into the THAP molecule deactivated unsubstituted NH groups in compounds **24** and **25**, thereby raising the proton acidity. Further on, we examined an option of using already partially acetylated THAP (compounds **24** and **25**) as the starting compound (Figure 7).

The resultant compound was characterized by physicochemical analytical methods. The IR spectrum had no absorption bands near NH stretching vibrations (3380–3200 cm^−1^), while the ^1^H NMR had only a singlet at 2.37 ppm typical of the acetyl CH_3_. Thus, the IR and NMR spectra evidence no protons in amino groups at positions 2, 4, 6, 8, 9, and 11 of the propellane structure and the formation of **26**.

Given the capabilities of the acetyl group to be substituted by the nitro group, it was of interest to perform such studies also for propellanes **24**–**26**. These compounds have free amines and acetyl-protected amines that can undergo both direct nitration and nitrolysis across the acetyl groups.

We detected no products from the further detachment of the acetyl group when compound **25** was treated with sulfuric acid, which is probably attributed to water being present in the system (Figure 8). Deacetylation and nitration were not observed by treating **25** with mixed nitric acid/acetic anhydride.

Similar manipulations with the other acetyl derivatives **24** and **26** in different media also gave a negative result. A temperature rise of the reaction mixture resulted in destruction of the THAP acetyl derivatives.

In the case mixed HNO_3_/(CH_3_CO)_2_O was used with product **24**, the analysis of the nitration products of **24** with mixed nitric acid/acetic anhydride showed that the reaction proceeded to furnish 3,7,10-trioxo-2,6-diacetyl-9-nitro-2,4,6,8,9,11-hexaaza[3.3.3]propellane **27** (Figure 9). A similar process was observed when compound **24** was treated with mixed acid and nitric acid.

### Quantum Chemical Prediction of 3,7,10-Trioxo-hexaaza[3.3.3]propellane and Its N-Nitro and N-Acetyl Derivatives

The structures of 3,7,10-trioxo-2,4,6,8,9,11-hexaaza[3.3.3]propellane **19** and its nitro- and acetyl derivatives were evaluated from quantum chemical predictions using the density functional theory (DFT). These predictions demonstrate that 3,7,10-trioxo-2,4,6,8,9,11-hexaaza[3.3.3]propellane **19** in the gas phase has two high-symmetry stable conformations: **19a** with C_3h_ symmetry and **19b** with D_6_ symmetry (Figure 5).

To evaluate the stability of molecules from the mono- to hexa-substitution, a dependence was built for the calculated heats of formation of compounds **21–26** (Figure 6 and Figure 7). A conformational analysis was preliminarily performed [22]. The data (Figure 6) refer to the most stable conformers of compounds **19, 21–22** and **24–26**. For comparison, we checked trinitrotoluene (TNT), hexanitrobenzene (HNB), CL-20, tetranitroglycoluril and reduced THAP with a different number of nitro groups (hexazapropellane, HAP).

According to the predictions, the successive insertion of nitro groups enhances the heats of formation of the resulting derivatives, whereas this tendency is opposite for the acetyl groups. It turned out that the hexaazanitro derivative of THAP is comparable in thermodynamic stability to hexanitrobenzene, while the reduced propellane with six nitro groups is comparable to CL-20.

A distinguishing feature of the constitution of propellane **19** is the elongated C1–C5 bond whose length in conformer **19a** is 1.59 Å. Curious is the behavior of the length of this bond when compound **19** is successively substituted by the nitro- and acetyl groups (Figure 7). The accumulation of the acetyl groups induces a monotonic shortening of the bond through to the regular length C(sp^3^)-C(sp^3^). In the case of the nitro groups, the tendency is more complicated: shortening first and then a significant elongation to 1.67 Å for six nitro groups in 3,7,10-trioxo-2,4,6,8,9,11-hexanitro-2,4,6,8,9,11-hexaaza[3.3.3]propellane (HNTHAP). Moreover, in the case of the nitro groups, the N-NO_2_ bonds considerably elongate from 1.43 Å in mononitro derivative **21** to 1.56 Å in HNTHAP. For the acetyl groups, the latter tendency is not pronounced.

It is hard to explain those patterns described above. It is evident that the matter is not (or not only) in steric factors because the acetyl group is more bulky [23]. It can be hypothesized that the loosening of all the strained bonds (propellane C1–C5 and N-NO_2_ bonds) in compound HNTHAP is the implication of its instability.

The replacement of the C=O group by the CH_2_ moiety (2,4,6,8,9,11-hexaaza[3.3.3]propellane, HAP) (Figure 8) results in a structural non-rigidness manifesting itself in a great number of stable conformations. For instance, the conformational search with subsequent optimization at the DFT level yielded 22 conformers in a range of 11 kcal/mol. A distinguishing feature of these conformers (and another manifestation of the structural non-rigidness) is the high variability in the C1–C5 bond length from 1.55 to 1.63 Å. A similar picture was also observed for the nitro derivatives of HAP.

Thus, based on the quantum chemical predictions, we gave a rationale for the formation of nitro- and acetyl derivatives of THAP. The instability of 2,4,6,8,9,11-hexanitro-THAP (HNTHAP), which exothermally decomposes when reacted with some solvents (DMSO, DMF, acetone), is on a par with the predicted high endothermic formation of this compound and is also explained by the loosening of the bonds within the molecule. The reduced THAP, hexaazapropellane, has shorter bonds, which will probably allow the synthesis of its hexanitro derivative in the future. The successful synthesis of the hexaacetyl THAP is explained by the fact that the molecule acquires additional stability when extra acetyl groups are inserted.

## 4. Conclusions

It was discovered herein that when 3,7,10-trioxo-2,4,6,8,9,11-hexaaza[3.3.3]propellane was nitrated with nitric acid, mixed nitric/sulfuric acids, and nitric anhydride, a competing lactam–lactim rearrangement took place, which was aided by low temperatures of the systems (nitric acid or mixed nitric/sulfuric acids). As the temperature was raised, the mono- and dinitro substitution products were shown to prevail. This study found that the direct acetylation of 3,7,10-trioxo-2,4,6,8,9,11-hexaaza[3.3.3]propellane occurs only in acetic anhydride to furnish di- and triacetyl derivatives. The synthesis of hexaacetyl-substituted THAP requires further acetylation of partially substituted derivatives of THAP with acetyl chloride. In addition, the acetyl derivatives were shown to be instable in acidic medium and to undergo deacetylation also when nitrated.

## Data Availability

Not applicable.

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
