# Peer review of "Synthesis of Nitro- and Acetyl Derivatives of 3,7,10-Trioxo-2,4,6,8,9,11-hexaaza[3.3.3]propellane"

_materials, 2022, doi:10.3390/ma15238320_

Round 1
Reviewer 1 Report
The nitration and acylation reactions of 3,7,10-trioxo-2,4,6,8,9,11-hexaaza[3.3.3] propellane were systematic studied in this paper. The two competing reactions during nitration were found, and investigating and obtaining the formation conditions of acylation derived products and the unstable nature of acetyl derivatives in acidic media. The results of this study should be of some reference value for the design and optimization of synthetic reaction conditions related to novel high-energy explosives.
The main problems that exist with papers are as follows:
1. Partial matter structural formulae are not sufficiently clear, making it difficult to discern differences in molecular structure. As compounds 20, 21, 22, the chemical bond identification in the structure diagram is unclear and needs to be further refined clearly.
2. It should be further stated why spiroalkyl heterocycles were used as nitrification precursors and what are the structural advantages of this azacycle over the hexaazaheterolignans. Only three data sets, predicted oxygen balance, density, and burst speed, are listed for selection with slight inadequacy.
3. It is suggested that the trend of the effect of the introduction of nitro group on the bond length should be further research analysis, and the speculation made according to the uncertain results cannot be taken as a final conclusion.
4. It is recommended that full text figures be presented in a uniform format with tables. The size of the structural formula is not consistent among multiple figures in the text (e.g., scheme 5 and Figure 7).
Author Response
The authors' response to reviewer 1 has been uploaded as a PDF file.

Reviewer 2 Report
Reference paper: materials-2012423
The manuscript reported the development of new energetic compounds that contain energetic explosophoric units based on the nitration reactions of THAP using various nitrating agents. Numerous energetic compounds were successfully prepared and fully characterized through thermal analysis, NMR, MS and IR. Interesting features have been obtained for which some compounds can be considered as potential energetic candidates for the next generation of energetic formulations. To complete the experimental investigations, the authors carried out a series of theoretical calculations to well understand the structure-property relationship. The paper is of interest to many researchers. The work has apparently been well executed. Perhaps the best part of this paper is the thoroughness of the approach. The authors do a good job at investigating details concerning the preparation of the next generation of energetic ingredients. The literature search is very thorough. I believe this paper is worth to be published in Materials.
Before publication, this referee suggests the authors to address the following points:
1- While the English grammar is acceptable, a thorough read-through would improve the paper.
2- Some important results should be mentioned in the abstract and not just introduce some the trivial information.
3- Introduction should be rewritten again to show the importance of the work and the gap that would be filled.
4- I suggest to move some important data from supplementary materials to the main document such as the thermal analysis results.
5- The authors have to determine the sensitivity of the developed compounds toward impact and friction.
6- The estimation of the performance parameters should be carried out.
7- Do the software products employed to perform the theoretical calculations required any experimental data/parameters/properties? If yes, what are these data/parameters/properties? Are them so effective than Gaussian 9?
8- Are they prone to be scaled up in the near future using the available readiness technologies?
Author Response
The authors' response to reviewer 2 has been uploaded as a PDF file.

Reviewer 3 Report
The authors conducted an elaborated analysis of the nitration of THAP by different nitrating agents. This paper is well written and organized. The work is very thorough and complete with the supplementary materials.
This is a very nice paper and should be published with very minor edits, see below.
1 The fonts in Figure 2 are too small.
Author Response
The authors' response to reviewer 3 has been uploaded as a PDF file.

Round 2
Reviewer 2 Report
The authors have revised the manuscript according to the reviewer comments, and the paper can be acceptable for publication in its present form.